# National audit of pressure ulcers and incontinence-associated dermatitis in hospitals across Wales: a cross-sectional study

Michael Clark,[1] Martin J Semple,[2] Nicola Ivins,[1] Kirsten Mahoney,[1,3] Keith Harding[4]

## ABSTRACT

**Objective** The Chief Nurse National Health Service Wales initiated a national survey of acute and community hospital patients in Wales to identify the prevalence of pressure ulcers and incontinence-associated dermatitis.

**Methods** Teams of two nurses working independently assessed the skin of each inpatient who consented to having their skin observed.

**Results** Over 28 September 2015 to 2nd October 2015, 8365 patients were assessed across 66 hospitals with 748 (8.9%) found to have pressure ulcers. Not all patients had their skin inspected with all mental health patients exempt from this part of the audit along with others who did not consent or were too ill. Of the patients with pressure ulcers, 593 (79.3%) had their skin inspected with 158 new pressure ulcers encountered that were not known to ward staff, while 152 pressure ulcers were incorrectly categorised by the ward teams. Incontinence-associated dermatitis was encountered in 360 patients (4.3%), while medical device-related pressure ulcers were rare (n=33). The support surfaces used while patients were in bed were also recorded to provide a baseline against which future changes in equipment procurement could be assessed. The presence of other wounds was also recorded with 2537 (30.3%) of all hospital patients having one or more skin wounds.

**Conclusions** This survey has demonstrated that although complex, it is feasible to undertake national surveys of pressure ulcers, incontinence-associated dermatitis and other wounds providing comprehensive and accurate data to help plan improvements in wound care across Wales.

## Strengths and limitations of this study

► This study identified the number of patients in hospital in Wales with pressure ulcers or incontinence-associated dermatitis.

► Visual inspection of patients' skin was undertaken to obtain accurate data on the prevalence of pressure ulcers and incontinence-associated dermatitis. Not all patients were able to participate in this skin inspection with the potential for inaccuracy in reporting where the skin was not observed.

► New pressure ulcers were identified during the audit, while other pressure ulcers had been incorrectly classified. These findings indicate that there is room for improvement in how nurses report pressure ulcers within Wales.

► All bar one Health Board used the Waterlow scale to assess pressure ulcer risk, judgements of low, medium and high risk may not be comparable between the six Health Boards that used Waterlow and the single Health Board that used an alternative risk assessment tool.

► While the presence of wounds other than pressure ulcers and incontinence-associated dermatitis was recorded, it was outside the scope of the audit for these wounds to be visually inspected.

[1]Welsh Wound Innovation Centre, Pontyclun, UK
[2]Welsh Government, Cardiff, UK
[3]Cardiff and Vale University Health Board, Cardiff, UK
[4]University of Cardiff, Cardiff, UK

**Correspondence to**
Professor Keith Harding; hardingkg@cf.ac.uk

## INTRODUCTION

The scale of the challenge of managing cutaneous wounds within the National Health Service (NHS) is becoming clearer through the exploration of databases capturing health information of people presenting with wounds at their general practitioner (GP) surgery.[1 2] From two databases Secure Anonymised Information Linkage (SAIL) and The Health Improvement Network (THIN), conservative estimates of the expenditure on wounds approach £330 million annually in Wales[2] and between £4.5 and £5.1 billion across the UK.[1] Both databases emphasise that the costs of wound care are largely driven by the cost of providing care rather than the cost of wound care products with wound dressings consuming 2.9% of total cost of wound care,[2] while wound care products accounted for 13.9% of the costs of wound care in the THIN database.[1]

While the two databases hold data on patients presenting with wounds in GP surgeries, differences exist between the two databases. For example, SAIL covers 41% of GP surgeries in Wales, while THIN holds data from 5.7% of GP surgeries across the UK, perhaps indicating that SAIL has deeper cover of a regional population, while THIN gives an impression of the range of care delivered

across the entire UK. There were also differences in the predominant wound aetiology identified within the two databases—with diabetic foot ulcers comprising 68% of the wounds identified by Phillips *et al*,[2] while leg ulcers of varying aetiology formed one-third of the wounds reported by Guest *et al*.[1] Pressure ulcers formed only 7% of the wounds reported from the THIN database but were one of the six main drivers of expenditure in the SAIL database along with diabetic foot ulcers, leg ulcers, foot ulcers, varicose eczema and postoperative wound care with these six wound aetiologies accounting for 93% of the costs of wound care. Previous estimates of the costs of wound care have also suggested that pressure ulcers are less common than wounds such as leg ulcers but cost more to treat than other chronic wounds.[3]

Over the years, there have been numerous attempts in the UK to specify how many pressure ulcers are present among hospital patients. Many of these epidemiological studies have been limited to single or small clusters of hospitals with data dependent on the recall of mainly nursing staff as to the number of patients with pressure ulcers present in the wards (eg, Clark and Cullum[4]). Over the past 16, years there has been a growing use of a robust methodology for reporting pressure ulcers developed by the European Pressure Ulcer Advisory Panel (EPUAP).[5] This approach depends on the independent assessment of the skin of each patient by two qualified nurses and is accepted to be time-consuming but leads to an accurate report of the number of patients with pressure ulcers.[6] While other territories have used the EPUAP method across a wide range of hospitals and other care locations[7–9] within the UK, the EPUAP method has been used to report pressure ulcer prevalence within orthopaedic units and a sample of community hospitals across Wales[10] with 13.9% (orthopaedic) and 26.7% community hospital patients having pressure ulcers. This paper reports the use of the EPUAP method of collecting accurate information on the number of patients with pressure ulcers across all hospitals within Wales providing the first national quantification of the size of the pressure ulcer population within any of the UK devolved nations. The survey was initiated by the Chief Nurse's Office, Welsh Government, and its primary objectives were to identify the number of patients with pressure ulcers and incontinence-associated dermatitis (IAD) in Welsh hospitals and the extent of misclassification of these two wound aetiologies. Secondary objectives were to record mattress provision to patients vulnerable to pressure ulcers or those with established pressure ulcers to explore whether support surfaces were appropriately allocated to patients and to record the presence of other wound aetiologies to identify the overall burden of wounds to hospital patients in Wales.

## METHODS

This was a cross-sectional survey of patients within hospitals in Wales found to have pressure ulcers or IAD. The survey is reported compliant with the STROBE checklist for cross-sectional studies.[11]

The NHS in Wales is divided into seven geographic Heath Boards covering 20 761 km$^2$ that provide primary and secondary care to their population along with three NHS Trusts (Velindre NHS Trust, Public Health Wales NHS Trust and the Welsh Ambulance Service NHS Trust). All Health Boards and Velindre NHS Trust (specialist cancer and blood services) participated in the audit with the audit methodology discussed and agreed both at director of nursing level and among the tissue viability nurses within all organisations. Preparation for the audit required almost 12 months to plan with the project led by the Welsh Wound Innovation Centre (WWIC) working with the Lead for Patient Safety and Patient Experience (Chief Nurse's Office). Detailed discussions occurred between each Health Board and WWIC to determine the scope of the audit within each Health Board with several issues arising, such as patient consent, independent skin assessment methods, capacity to undertake the audit and the decision to include wound aetiologies other than pressure ulcers and IAD. It was agreed that WWIC would provide staff to assist the audit where required and that while only pressure ulcers and IAD would be visually inspected, other wound aetiologies would be reported but not visually seen by the audit team. While this audit primarily aimed at collecting data in acute and district general hospitals, a number of community hospitals were also included, while Powys Teaching Health Board audited all patients within its community hospitals (this Health Board is community based having no acute or district general hospitals).

The audit followed the methods established by the EPUAP,[5] where two independent experienced nurses inspected the skin from head to toe of all patients who gave consent with all pressure ulcers identified and classified as either category I, II, III or IV injuries.[12] Category I marks damage of unbroken skin, category II a superficial wound with categories III and IV marking full-thickness skin and soft tissue loss. Additional categories of pressure damage included suspected deep tissue injury and unstageable wounds.[12] The National Pressure Ulcer Advisory Panel/EPUAP/Pan Pacific Pressure Injury Alliance[12] pressure ulcer classification (including suspected deep tissue injury and unstageable wounds) was in use across all Welsh Health Boards and familiar to all clinical staff who participated in the audit. All differences between the categories assigned to the encountered pressure ulcers by the two assessors were resolved through discussion. All assessors who classified wounds were experienced wound care/tissue viability nurses competent to assess pressure ulcers and IAD.

IAD was defined as skin wounds caused by urine and/or faeces and perspiration, which is in continuous contact with intact skin of the perineum, buttocks, groins, inner thighs, natal cleft, skin folds and where skin is in contact with skin.[13] This definition of IAD was used across all Health Boards and was familiar to the clinical staff who participated in the audit.

Patient demographic information was collected from ward records with gender recorded as male or female and age gathered in interval bands (for example 80–89 years) rather than as specific ages.

Vulnerability to pressure ulcer development was assessed using the pressure ulcer risk assessment tool used within each Health Board with all bar one using the Waterlow risk assessment tool[14] with Waterlow scores of between 10 and 14 used to mark low risk of pressure ulcer development, 15–19 is medium risk and 20 and above is high risk of pressure ulcer development. One Health Board used an alternative risk assessment tool—the Pressure Sore Prediction Score (PSPS).[15] In this system, scores of 6–9 marked low risk, 10 and 11 marked medium risk and 12–16 marked high risk. This audit did not calculate a pressure ulcer risk assessment score for each patient relying on the calculations of patient vulnerability to developing pressure ulcers reported on each ward. This limitation is raised in the discussion.

Secondary outcomes of the audit across Welsh hospitals were to explore the appropriateness of pressure redistributing mattresses to patients with, or at risk of, pressure ulcers. Data were collected using the mattress brand name, which was checked against manufacturers' descriptions to identify each mattress as being either a foam mattress, hybrid foam and air, other static mattress, low air loss product/specialty bed, dynamic overlay or dynamic replacement mattress.

The final outcome measure identified the presence of wounds other than pressure ulcers or IAD. This was undertaken to identify the total burden of wounds to hospitals in Wales. No visual examination of these other wound aetiologies was possible, and the accuracy of reporting was unknown.

There were a number of differences in the data collected during the audit from the Minimum Data Set proposed by the EPUAP. The teams of two nurses who collected pressure ulcer data were supplemented by a data recorder, including volunteers from the staff of wound care companies and staff and students who were keen to help whose role was to enter data directly into an electronic database. The EPUAP Minimum Data Set also recorded whether patients were manually repositioned and whether the patient had been provided with a powered or non-powered mattress and/or seat cushion. The Welsh pressure ulcer audit collected greater detail around mattress provision but did not record cushion use or repositioning.

Survey participants were all inpatients in each hospital present at midnight on the night before the audit. In each clinical area, a date and time for the audit visit was agreed with clinical staff with the ward staff completing a form reporting pressure ulcers present from all inpatients at midnight of the night before the audit visit. Two clinicians and a data collector visited each clinical area; the clinicians sought consent from each inpatient to have their skin examined if consent was provided. No skin inspections were undertaken among mental health patients; this was a limitation agreed during the planning of the audit to minimise issues around seeking consent where capacity to provide consent may be reduced. If the skin inspection identified a pressure ulcer or IAD that was not recorded on the form completed by the ward staff, then these wounds were also recorded. All data were captured on paper (completed by ward staff) and electronically using iPads (Apple UK) with hardware and information technology support provided by Medstrom, a supplier of pressure redistributing equipment. The audit was conducted collaboratively across Wales bringing together the tissue viability nurses within each Health Board, WWIC clinical staff and a wide range of clinical and non-clinical staff from the wound care industry in Wales.

Medstrom, using a series of Excel (Microsoft USA) files, undertook initial data cleaning to remove duplicated data and identify missing data. Excel files containing data from single Health Boards were then circulated to each Health Board's tissue viability nurses to check the accuracy of the data gathered within their Health Board. WWIC then created an SPSS (V.23.0) data file bringing together all the data for analysis.

All data were reported using mean, SD and 95% CI where these summary measures could be calculated. Mean age could not be calculated given the collection of these data in banding intervals (eg, 80–89 years). No attempts were made to impute missing data with the numerator and denominator given for each point estimate. No attempt was made to correct for missing data with the numerator and denominator provided for each calculation.

### Governance
This clinical audit was approved by the director of nursing for each participating organisation, and no formal research ethics approvals were required.

## RESULTS
### Demographic data
The audit was undertaken during the period 28 September 2015 to 2 October 2015 with data collected on 8365 patients located across 66 hospitals. Of the patients surveyed, 4659/8365 (55.7%) were female and 3329/8365 (39.8%) were at least 80 years old. There were 4282 out of 8279 (51.7%) patients at a medium or high risk of developing pressure ulcers with the level of risk of 86 patients to pressure ulcer development unreported.

### Pressure ulcers
Seven hundred and forty-eight patients (748/8365; 8.9%, 95% CI 8.29% to 9.51%) had pressure ulcers with these wounds being either reported by ward staff and/or observed during the audit. These 748 patients had 907 pressure ulcers. It was not possible to compare the accuracy of the ward reports of pressure ulcers as in 2887/8065 (35.8%) no direct observation of the patient's

skin was undertaken. No skin inspection was undertaken for those patients with mental health problems (n=1004) due to issues around their capacity to consent to the skin inspection. Additionally, no skin inspection was undertaken if the patient did not provide consent (n=233), was too ill to consent (n=390) or was not present on the ward at the time of the audit visit (n=576). For 300 patients, the reason their skin was not inspected was unreported with 164 (54.7%) of these located in a single Health Board. The final 684 patients declined to have their skin inspected by the audit team.

### Pressure ulcers verified through observation of the skin by the audit teams

Of the 748 patients with pressure ulcers, visual verification of their most severe pressure ulcer and its classification was available in 593 (79.3%) cases. Patients with verified pressure ulcers tended to be female (n=319; 53.8%) with 330 (55.7%) over 80 years old; most (n=520; 87.7%) were at medium to high risk of developing pressure ulcers with only seven patients with verified pressure ulcers reported to be not vulnerable to pressure ulcer development. One of the seven had a medical device-related pressure ulcer; the reason for pressure ulcer development in the remaining six risk-free patients was unknown.

The majority of patients with verified pressure ulcers had a single pressure ulcer (n=493) with 32 patients having three or more pressure ulcers (maximum six pressure ulcers in a single patient). Of the 593 patients with verified pressure ulcers, 266 (n=44.8%) had been admitted to their current care location with their pressure ulcer(s) with 259 (43.7%) developing pressure ulcers postadmission. The origin of the verified pressure ulcers of 68 patients was unreported suggesting there is room for improvement in pressure ulcer reporting across Wales.

Table 1 details the severity and anatomical location of the most severe verified pressure ulcer, with category II pressure ulcers being the most common injury at each location. Severe pressure ulcers (categories III and IV, deep tissue injury and unstageable) were verified in 152/587 (25.9%) patients with the maximum severity of verified pressure ulcers among six patients unreported.

In 331 (55.8%) patients with verified pressure ulcers, the classification provided by the ward staff matched that of the audit teams covering 435 superficial pressure ulcers (categories I and II) and 152 severe pressure ulcers. One hundred and thirty-two (22.2%) patients were found to have pressure ulcers not reported by ward staff, while 124 (20.9%) patients had one or more pressure ulcers incorrectly classified by the ward team. Ward staff did not report 158 verified pressure ulcers and reported incorrect classification for 152 verified pressure ulcers. No patient reported to have a pressure ulcer was found to be pressure ulcer free on inspection of their skin.

For 26 patients, their verified pressure ulcers were caused through contact with medical devices. Black *et al*[16] reported that within a single US acute care hospital, 34.5% of all hospital-acquired pressure ulcers were medical device related; in the present audit, there were fewer patients with hospital-acquired medical device-related pressure ulcers (20/217; 9.2%). This finding may reflect that there is a need for further education of staff to recognise pressure ulcers caused by medical devices.

### Pressure ulcers reported by ward staff but not verified by skin observation by the audit teams

One hundred and fifty-five patients (20.7%) had reported pressure ulcers that were not verified by the audit teams (table 2). This group were broadly similar to patients with verified pressure ulcers in terms of age and their vulnerability to developing pressure ulcers (86 (55.5%) over 80 years old and 138 (90.2%) at medium to high risk of developing pressure ulcers). However, most patients with reported pressure ulcers were male (n=82; 52.9%).

Most patients with reported pressure ulcers tended to have a single pressure ulcer (n=139; 89.7%) with two patients each having three reported pressure ulcers. Most patients with reported pressure ulcers were admitted with their wound (n=85; 54.8%) with 78 (50.3%) patients developing their pressure ulcer(s) postadmission. The origin of the pressure ulcers of 12 patients with reported pressure ulcers was unknown.

Table 2 details the severity and anatomical location of the most severe reported pressure ulcers with category II pressure ulcers the most commonly reported form of pressure ulcer. The anatomical location of the most severe pressure ulcers of two patients was unreported. Few deep tissue injuries and unstageable pressure ulcers were reported by ward staff with two and six patients,

**Table 1** Severity of the most severe verified pressure ulcer per patient by anatomical location of these wounds

| Anatomical location | Category of pressure ulcer | | | | | | | |
|---|---|---|---|---|---|---|---|---|
| | I | II | III | IV | Deep tissue injury | Unstageable | Unknown | Total |
| Sacrum | 70 | 93 | 26 | 12 | 4 | 8 | 2 | 215 |
| Heel | 44 | 60 | 22 | 7 | 7 | 18 | 3 | 161 |
| Buttock | 32 | 59 | 6 | 4 | 0 | 4 | 0 | 105 |
| Other | 23 | 50 | 12 | 3 | 4 | 15 | 1 | 108 |
| Total | 169 | 262 | 66 | 26 | 15 | 45 | 6 | 589 |

Anatomical location of four patients' most severe pressure ulcer unreported.

**Table 2** Severity of reported but non-verified most severe pressure ulcer per patient by anatomical location of these wounds

| Anatomical location | Category of pressure ulcer | | | | | | | |
|---|---|---|---|---|---|---|---|---|
| | I | II | III | IV | Deep tissue injury | Unstageable | Unknown | Total |
| Sacrum | 22 | 35 | 5 | 5 | 0 | 2 | 3 | 72 |
| Heel | 11 | 9 | 3 | 3 | 2 | 3 | 1 | 32 |
| Buttock | 9 | 10 | 3 | 0 | 0 | 1 | 0 | 23 |
| Other | 7 | 12 | 4 | 2 | 0 | 0 | 1 | 26 |
| Total | 49 | 66 | 15 | 10 | 2 | 6 | 5 | 153 |

Anatomical location of the most severe pressure ulcers in two patients unreported.

respectively, reported to have these forms of pressure ulcer. Seven patients had reported pressure ulcers that were stated to have been caused through contact with medical devices.

### Pressure redistributing support surfaces

The use of a pressure redistributing mattress is one element of pressure ulcer prevention; the survey enabled collection of data on the current use of pressure redistributing mattresses to allow assessment whether patients were provided with appropriate surfaces based on either their vulnerability to pressure ulcer development or the severity of their existing pressure ulcers. Support surface appropriateness was based on current recommendations for support surface use within each Health Board. A wide range of patient support surfaces were encountered during the audit for use while the patient rests in bed. These were simplified into six categories: foam mattress, other static mattresses and overlays, low air loss/specialty bed, mattress with static and dynamic capability (hybrid mattress), dynamic mattress overlays and dynamic mattress replacement systems.

Tables 3 and 4 describe the allocation of pressure redistributing support surfaces by vulnerability to pressure ulcer development and severity of pressure ulcer respectively. Six hundred and sixty-three (32.4%) patients at high risk of pressure ulcer development rested on foam mattresses, while 36 patients considered not to be at risk were allocated dynamic mattress replacements. From table 4, eight people with 'unstageable' (full thickness)

pressure ulcers were allocated foam mattresses, while 49 people with category I pressure ulcers were nursed on dynamic replacement mattresses indicating that even if pressure-redistributing support surfaces are available, they are not always under the correct patient.

The mattress allocated to 1308 (15.6%) patients was unreported with 1035 unreported support surfaces located in a single Health Board. This finding indicates that even with careful planning of a national audit, there are occasions when local data collectors do not comply with the audit protocol indicating a requirement for greater emphasis preaudit on the need to collect all data.

### Incontinence-associated dermatitis

There were 306 patients with verified IAD and 56 with reported but unverified IAD giving a total of 362 (4.3%; 95% CI 3.91% to 4.79%). No IAD had been recorded by ward staff as a pressure ulcer, although six pressure ulcers had been recorded by ward staff as being IAD.

### Other wounds

While the survey was intended to capture the number of people with pressure ulcers and IAD, other wound aetiologies were recorded but not verified through observation by the audit teams. Of the 8365 patients surveyed, 2537 (30.3%) either had a pressure ulcer, IAD or another wound. There were 56 patients having both a pressure ulcer and IAD. Ninety-four patients had either pressure ulcers or IAD along with another wound. Table 5 lists the most common other wound aetiologies recorded; a

**Table 3** Allocation of mattresses by level of vulnerability to pressure ulcer development

| Product type | Risk of pressure ulcer development | | | | |
|---|---|---|---|---|---|
| | No risk | Low | Medium | High | Total |
| Foam mattress | 1198 | 1665 | 862 | 663 | 4388 |
| Other static mattress/overlay | 40 | 223 | 299 | 533 | 1095 |
| Low air loss/specialty bed | 0 | 1 | 7 | 15 | 23 |
| Hybrid product | 2 | 7 | 22 | 70 | 101 |
| Dynamic overlay | 4 | 23 | 40 | 119 | 186 |
| Dynamic replacement | 36 | 161 | 323 | 644 | 1164 |
| Total | 1280 | 2080 | 1553 | 2044 | 6957 |

The risk of developing pressure ulcers was unknown in 38 patients with a reported support surface, while 62 patients rested on unspecified dynamic mattresses, divan beds, chairs or trolleys.

**Table 4** The allocation of pressure redistributing mattresses by the severity of pressure ulcer.

| Product type | Pressure ulcer classification | | | | | | |
| | I | II | III | IV | Deep tissue injury | Unstageable | Unknown |
| --- | --- | --- | --- | --- | --- | --- | --- |
| Foam mattress | 68 | 84 | 11 | 1 | 1 | 8 | 7 |
| Other static mattress/overlay | 58 | 80 | 10 | 9 | 1 | 5 | 1 |
| Low air loss | 0 | 4 | 0 | 3 | 0 | 0 | 0 |
| Hybrid product | 7 | 11 | 4 | 6 | 0 | 2 | 0 |
| Dynamic overlay | 8 | 14 | 9 | 1 | 0 | 2 | 0 |
| Dynamic replacement | 49 | 93 | 26 | 12 | 8 | 22 | 4 |
| Total | 190 | 286 | 60 | 32 | 10 | 39 | 12 |

wide range of other wound aetiologies and anatomical locations where wounds were reported to occur were reported; however, each affected fewer than 30 patients.

## DISCUSSION

This pressure ulcer audit identified 748/8365 (8.9%) patients with a pressure ulcer(s) within acute and community hospitals across Wales. The survey methodology aimed to visually inspect the skin of each patient, other than mental health patients, to allow verification of ward staff-reported pressure ulcers. However, skin was only inspected in 5178 patients with 1004 mental health patients and 1883 patients with no skin inspection for a variety of reasons including actively declining to have their skin inspected (n=684). Other cross-sectional pressure ulcer prevalence surveys have reported patient exclusions, for example, Bredesen et al[8] reported 125 patients were excluded from a potential sample of 1334 patients in one region of Norway. Other pressure ulcer audits[9] did not report patient exclusions potentially as relative assent was gained for participation. The results of the Welsh pressure ulcer audit have been presented in terms of verified and reported pressure ulcers; a mechanism that may be helpful in future pressure ulcer surveys where direct observation of the skin is the goal but where

staff reports of the occurrence of wounds may have to be used where skin assessment is not possible.

The anatomical location and severity of the encountered pressure ulcers reflected those seen in other surveys[8 9] with the majority being superficial pressure ulcers located at the sacrum, buttocks and heels. There were few differences in the nature of the pressure ulcers that were verified through observation or reported by ward staff with observations of deep tissue injury and unstageable wounds rarer where pressure ulcers were reported perhaps indicating less sophistication among ward nurses to the subtleties of pressure ulcer categorisation compared with wound healing and tissue viability specialist nurses who tended to form the majority of the audit teams.

Where verification of pressure ulcers was undertaken, the majority of ward based and audit team identification and classification of the encountered pressure ulcers agreed (n=331 patients of the 593 with verified pressure ulcers; 55.8%). However, 158 new pressure ulcers were found by the audit teams with a further 152 pressure ulcers incorrectly categorised by the ward teams; these errors in identification and classification affected 310/907 (34.2%) of the pressure ulcers found during the audit suggesting a need for further education and

**Table 5** Most common other wound aetiologies recorded in the 2015 wound audit. In each case, the denominator was 8365 patients.

| Wound aetiology | Number of patients (mean prevalence, 95% CI) |
| --- | --- |
| Closed surgical wound | 841 (10.05%; 9.41% to 10.69%) |
| Other surgical wound | 55 (0.66%; 0.49% to 0.83%) |
| Infected surgical wound | 43 (0.51%; 0.36% to 0.66%) |
| Dehisced surgical wound | 35 (0.42%; 0.28% to 0.56%) |
| Skin tear | 215 (2.57%; 2.23% to 2.91%) |
| Leg ulcer (no differential diagnosis) | 196 (2.34%; 2.02% to 2.66%) |
| Diabetic foot ulcer | 56 (0.67%; 0.5% to 0.84%) |
| Traumatic wound | 40 (0.48%; 0.33% to 0.63%) |
| Lymphoedema | 37 (0.44%; 0.3% to 0.58%) |
| Wound diagnosis or location unknown | 115 (1.37%; 1.12% to 1.62%) |

training on pressure ulcer recognition and classification while justifying the investment in time required to undertake a detailed audit across a wide range of hospital sites.

The survey also identified 306 verified IAD with a further 56 reported giving a prevalence of 4.3% across the inpatient population of Wales. Reports of the prevalence of IAD are rare in the literature and tend to focus on individual organisations[17][18] suggesting that the present survey across the entire inpatient population of Wales contributes to the growing discussion around IAD. Few misclassifications were observed between IAD and pressure ulcers suggesting a sound level of knowledge related to the differential diagnosis of these injuries.

Medical device-related pressure ulcers were relatively rarely encountered with 20 of 217 (9.2%) pressure ulcers that developed in hospital caused by a medical device compared with 34.5% of incident pressure ulcers being caused by medical devices in one US-based study.[16] There may be value in training NHS staff to recognise medical device-related pressure ulcers to ensure that this group of avoidable cases of pressure damage can be prevented.

Pressure redistributing mattresses were recorded across all bar one Health Board with apparent discrepancies between mattress allocation and either the degree of vulnerability of patients to developing pressure ulcers or the severity of established pressure ulcers. Mattress allocation may be the end-product of a complex process of order and supply of these devices,[19] and it is feasible that a cross-sectional survey may not adequately reflect the processes involved in mattress allocation. Regardless of this, the results from the survey suggests a need for improvement in how mattress stocks are allocated to patients.

The audit was intended to verify the number of inpatients with pressure ulcers and IAD, although the number and type of other wounds was recorded with no verification. Taken collectively, the survey indicated that over 30% (n=2537; 30.3%) of all inpatients across Wales had one or more wounds highlighting the common occurrence of wounds and the requirement for improvements in both prevention and treatment to reduce the burden placed by wound care on the Welsh NHS. Of the wounds reported, there were 841 closed surgical wounds; the inclusion of these might be criticised given that these were closed wounds; excluding closed surgical wounds, the burden of wounds across Welsh inpatients fell to 1696/8365 (20.3%) equivalent to one in every five inpatients with an open wound. The justifications for the inclusion of closed surgical wounds were the finding that postoperative wound care was one of the six main drivers of the cost of wound care in Wales,[2] and surgical site infections (SSIs) remain common ranging from 0.3 SSI per 1000 inpatient days (knee prostheses) to 8.2 per 1000 patient days (large bowel).[20]

This survey was commissioned by the Chief Nurse's Office within Welsh Government, and the results have been disseminated from the Chief Nurse to the individual Health Boards and NHS Trusts in Wales. That the planning of the survey and the dissemination of the results have been managed from within government has helped accelerate changes arising from the survey results. These changes will impact on nursing staff knowledge and training through the creation of an educational module to be completed by all nurses in Wales that guides them to a better understanding of pressure ulcer identification and classification and a review of formal wound care education at undergraduate degree level including what is taught during clinical placements. Improvements in procurement of pressure redistributing support surfaces will be gained through the closer integration of the WWIC with procurement processes and strategies. This audit has shown that with the appropriate support within the leadership of the NHS in Wales, it is possible to generate national level data to justify and reinforce the need for changes in education, procurement and practice.

There were limitations to the reported survey. One of these limitations was the inability to assess the risk of all patients to pressure ulcer development using a common risk assessment tool. While the Waterlow scale was used in all bar one Health Board, the comparability of risk assessments recorded using Waterlow and PSPS data is unclear, and future surveys may wish to consider calculating a pressure ulcer risk assessment score for all patients. The use of the rigorous EPUAP methodology with independent assessment of the skin by two observers increased the planning and workload of the survey. Kottner et al[21] have reported on the accuracy of pressure ulcer identification and classification based on a single or two data collectors with no apparent difference in accuracy where only a single data collector gathered data. It may be possible to reduce the planning and complexity of national audits if a single data collector was used to collect data rather than pairs of data collectors for future audits.

Undertaking this complex audit has enabled Welsh Government and NHS Wales to gain an increased appreciation of the need to capture comprehensive, accurate data on wound occurrence and outcomes to ensure continuous service improvements are achieved, and our findings provide support to establish the extent of other areas of wound care suitable for improvement while recognising that large-scale studies such as this are difficult but feasible. While this work was commissioned to provide insight into pressure ulcers and IAD in Welsh hospital patients, the survey does provide guidance towards the planning, execution and reporting of national wound audits regardless of geographical location. Central planning of the survey through the Chief Nurse's Office facilitated agreement across the data to be collected and the patient groups to be included or excluded (eg, mental health patients). It is possible that access to central planning will be a key requirement for the undertaking of national wound audits. The separation of the collected data into verified (through observation) and recorded wounds gives a mechanism through which the robustness of the data gathered during a national audit can be identified even where the goal was to be able

to verify all wounds through direct observation of the skin with this goal not met in many cases in the present survey.

**Acknowledgements** This audit would not have been possible but for the support from the directors of nursing and the willing participation of so many people especially from the All Wales Tissue Viability Nurse Forum and the WWIC. We thank Medstrom for their IT support for this audit and all of the data input staff drawn from a wide range of commercial organisations supplying Wales with wound care products.

**Contributors** MC was involved in the planning of the audit with the Chief Nurse's Office, analysed the data and prepared the drafts and final version of the manuscript. MJS coordinated planning of the audit, gaining permission from all participating organisations and reviewed the final manuscript. KH, NI and KM suggested analyses, reviewed the results and edited the final manuscript.

**Competing interests** None declared.

**Provenance and peer review** Not commissioned; externally peer reviewed.

**Data sharing statement** No additional data are available.

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
