## [Reviewer comments · BMJ Open]

ARTICLE DETAILS

TITLE (PROVISIONAL)	A national audit of pressure ulcers and incontinence associated dermatitis in hospitals across Wales: a cross-sectional study.
AUTHORS	Clark, Michael; Semple, Martin; Ivins, Nicola; Mahoney, Kirsten; Harding, Keith

VERSION 1 - REVIEW

REVIEWER	Jan Kottner Charité-Universitätsmedizin Berlin, Germany
REVIEW RETURNED	23-Dec-2016

GENERAL COMMENTS	Thank you very much for the invitation to review this manuscript. This authors present results of an epidemiological study of pressure ulcers and other lesions using a sound methodology. Accurate epidemiological figures are important to evaluate the burden and importance of pressure ulcers in different populations. Please find my comments below. (1) The population based approach of this work is a major strength. In order to make this manuscript more useful and powerful I urgently recommend to restructure the whole manuscript according to the STROBE statement. Addressing subsequently all STROBE ensures maximum transparency and that all details are provided to fully understand what was how conducted. Please add in the Methods a variables section (e.g. how was pressure ulcer risk measured and categorized, see later Table 3), the statistical analysis etc. (all STROBE items). Please also follow the statistical recommendations, e.g. provide 95% CI around the point estimates. (2) Introduction: While I would support all statements made it is a little bit unclear for the reader why exactly this research was needed. Please state the aims/objectives clearly especially with regard to all the results presented later (e.g. mattress use etc). (3) Introduction: Please provide the background information in the Introduction, e.g. how 'moisture lesions' were defined. What is the problem with diagnoses. (4) Results: Please provide the demographic data in sufficient detail including mean age and SD. Please provide the prevalence estimates with 95% CI and numerator and dominators in the tables 1 and 2. (5) Results: What exactly was the rationale for tables 3 and 4? Please explain before. (6) Results, table 5: Please provide 95% CIs and denominators.
--

	(7) Methods/results: Please consider a 2x2 table for displaying and calculating agreement/disagreement for documented PUs and PUs found during the survey. Same for moisture lesions. Among other this this would provide information about a possible detection bias. (8) Discussion: Please add the contribution of this manuscript to the field of pressure ulcer/wound care research and practice. At the moment the discussion largely focuses on Wales and NHS.
--	---

REVIEWER	Ronald Houwing Deventer Ziekenhuis the Netherlands
REVIEW RETURNED	09-Jan-2017

GENERAL COMMENTS	Comments on manuscript BMJ open -2016-015616 A national audit of pressure ulcers and incontinence associated dermatitis in hospitals across Wales. Comments from Houwing R.H. MD , Koopman E.S.M. R.N. Introduction: The reason for the initiative to conduct this study remains unclear, wich makes it unclear if the method complies with the goal of this study. Does the chief nurse wants to know the extent/cost of the problem? Does he want to know if patients are receiving adequate care? Does he want to know if there are differences between health care organisations? It appears to us that the primary reason is to get an idea of the magnitude of the problem. Why not use the Maastricht method of prevalence PU reporting. Introduction page 4. Our advice would be to use the term point prevalence throughout
---

the article.

Methods page 4

As moisture lesion is not a ICD-10 diagnosis, why not leave that out and replace that with the correct term incontinence-associated-dermatitis IAD?

How accurate are the data as 155 of 748 patients with PU have not been observed?

Several studies have shown that reported PU's do not reflect the reality because of missed diagnosis and wrong diagnoses.

Page 7

The origin of verified PU was under reported. Suggesting room for improvement is a conclusion not meant for the chapter results (also page 8). It might be better to list all conclusions in a separate paragraph.

Because the chosen method is very unusual, we would like to see that the method is clarified in a diagram with an overview of the study.

Page 8

What is meant with mental health. Only psychiatric disorders or are Alzheimer patients also excluded? As the latter are especially vulnerable.

Why is the number of patients whose skin was actually inspected so low. only 64,2% included?

The strength of this study could be heightened with a higher participation.

155 of the pu patients were not seen by the audit team.

Page 8

The category of PU is used of the latter EPUAP classification, included the Deep tissue injury and unstageable. (6 categories)

Contrary to the methods page 4, where 4 categories are described.

	Page 9 Moisture lesion. It is not possible to make a sharp differentiation between the so-called moisture lesion and pressure ulcers based on the clinical spectrum. The ward staff misclassified pressure ulcer for moisture lesions. Discussion page 10 In future pressure ulcer study it is not wise to use the same method as used in this study. Better not combine reported and observed PU in the same study. Diagnosis only by trained observers reporting only observed PU whether in point prevalence, preferably in period-incidence. Page 11 This study confirms the difficulty in classification of PU, in accordance to a lot of studies. My recommendation would be to limit the number of classifiers, so that time is won for nursing staff to focus on adequate preventive interventions instead of lengthy discussions and education on classification. Page 11 "The audit was intended to verify the number of in-patients with PU and moisture lesions". This is not mentioned in the introduction as a goal of this study. But still; what was the purpose of identifying the number of PU?
--	--

REVIEWER	Yufitriana Amir Universitas Riau, Indonesia
REVIEW RETURNED	18-Jan-2017

GENERAL COMMENTS	It is a good study about feasibility of national surveys of pressure ulcers. Abstract Please do not use abbreviation in the abstract. Please mention the reason of exclusion 1004 mental health patients
--

	in the methodology. Introduction Page 4: Please give some explanation about difficulties to conduct national surveys of pressure ulcers and add about national pressure ulcer studies in other countries such as The Netherlands, Austria, German, Japan. Page 5.,Vanderwee et al (2007). It should be(Vanderwee et al, 2007). Please explain more details about data collection procedure base on EPUAP mini data set survey and how to collect data in the large scale. Concerning number of raters, you can also this reference Kottner, J., Tannen, A., & Dassen, T. (2009). Hospital pressure ulcer prevalence rates and number of raters. Journal of Clinical Nursing, 18(11), 1550-1556. Page 4 line 54 add reference. Page 5 line 47. How about the other categories; unstageable, suspected deep tissue injury, and unknown? Mention them in the method. Page 5 page 48-49. Did the two assessor have training about PU categories? Are they competence to assess Pu wound and incontinence-associated dermatitis. Page 6 line 51. What do you mean with “governance”? Please mention about data analysis and research ethics (e.g. participant consent, ethics approval) Page 7 line 8. I do not understand “495.5 person days”. Please explain it. Page 7 ilne 18. How to calculate “none-low-medium/high risk of PU”. Please mention it in the methods. Page 45 “suggesting there is room for improvement...”. Mention this in the discussion. Page 8 Line 9-14. Mention this in the discussion, Table 1. Could you provide more detail about “other” anatomical location of PU? Page 10 line 2. Please maka a table about PU and incontinence-associated dermatitis. Please use up-date term for moisture lesion. Discussion Page 10 line 20. Please mention the reason of inclusion criteria Please add constrains/difficulties on conducting national measurement. Please add study limitation and conclusions.
--	---

REVIEWER	Catherine VanGilder Hill-Rom Clinical Research
	Employee of Hill-Rom, but no competing interests.
REVIEW RETURNED	01-Mar-2017

GENERAL COMMENTS	This is a very important paper and definitely needs to be published. I would like to see the authors clarify the 2 levels of analysis more clearly. What did the ward staff find, and then what did the reviewers find. Perhaps a schematic starting with all patients included, # PU patients and # Pressure ulcers. Additionally, there are many confusing run on sentences that should be addressed. Great work and we are very interested in seeing this move forward! The reviewer also provided a marked copy with additional comments. Please contact the publisher for full details.
---

VERSION 1 – AUTHOR RESPONSE

Reviewer 1 suggested we reconfigure the manuscript to make it compliant with the STROBE statement. This has been undertaken in the revision with changes to the title, introduction, methods, results and discussion to ensure the content is consistent with the reporting requirements of STROBE. We have also added text to the introduction to state the aims and objectives of the survey as recommended by reviewer 1. We collected age in ten year bands (for example 80 to 89 years) this precludes us from presenting mean age as suggested by reviewer 1. We have not added 95% confidence intervals to tables 1 and 2 as the two tables present the maximum severity of pressure ulcer per patient rather than estimates of prevalence. The titles of tables 1 and 2 have been amended to better explain the content of the tables. We have added rationale for the inclusion of Tables 3 and 4 within the introduction and methods. As suggested by reviewer 1 we have added 95% Confidence Intervals around the prevalence of wounds other than pressure ulcers and incontinence-associated dermatitis.

Reviewer 2 also commented upon the objectives of the audit and this has been clarified within the introduction. We would disagree that the EPUAP prevalence methodology is 'very unusual' having been reported in several publications (for example Vanderwee K, Clark M, Dealey C, Gunningberg L, Defloor T. Pressure ulcer prevalence in Europe: a pilot study. J. Eval. Clin.Pract. 2007; 13(2): 227-35.) but recognise that the Maastricht method has been used with success in many pressure ulcer surveys. We have not used the term 'point prevalence' in the manuscript given that the data was collected over the period of one week. Reviewer 2 commented that the number of patients where their skin was inspected was low – full details of the exclusions have been incorporated into the manuscript.

Three reviewers suggested adding 2x2 tables to illustrate the pressure ulcers reported by ward staff and those found during the survey. We have not included the 2x2 tables but have added explanatory text to highlight that we could not directly compare the ward reported pressure ulcers with the observations of the data collectors given the number of patients where skin assessment was not possible accounting for 35.8% of all patients included in the survey. Where skin assessments were performed, we have amended the results section to identify the number of patients and number of pressure ulcers that were found during observation that had not been identified by ward staff. We have also added comment that no patient reported to have a pressure ulcer was observed to be pressure ulcer free. We have included broader discussion of the contribution of the manuscript to pressure ulcer research along with discussion of the limitations of the survey.

Reviewer 3 invited us to include additional details of national pressure ulcer surveys however we have restricted our discussion to those national surveys that used the EPUAP data collection methodology. Other national surveys for example in Germany, Austria and the Netherlands used alternative data collection methods. We have included the reference Kottner et al 2009 within the discussion section of the manuscript as suggested by reviewer 3. The other suggestions made by reviewer 3 related to the description of pressure ulcer categories and risk assessment were covered when we amended the manuscript to make it STROBE compliant. Reviewer 4's comments addressed the inclusion of 2x2 tables and have been addressed above.

VERSION 2 – REVIEW

REVIEWER	Jan Kottner Charité-Universitätsmedizin Berlin, Germany
REVIEW RETURNED	07-Apr-2017

GENERAL COMMENTS	Thank you very much for doing the revisions. The manuscript has
---

	improved substantially.
--	-------------------------

REVIEWER	Catherine VanGilder Senior Manager, Global Clinical Research, Hill-Rom
REVIEW RETURNED	27-Apr-2017

GENERAL COMMENTS	The numbers just don't match up. I cannot follow the tables with the text. I am included a scanned pdf with my comments The reviewer also provided a marked copy with additional comments. Please contact the publisher for full details.
--

VERSION 2 – AUTHOR RESPONSE

Thank you for sending us additional comments from two reviewers upon our manuscript 'A national audit of pressure ulcers and incontinence associated dermatitis in hospitals across Wales: a cross-sectional study'.

We were pleased to see that Reviewer 1 found the manuscript much improved and that original reviewers 2 and 3 had no further comment. We thank reviewer 4 for their additional comments which we have addressed below and in the revised text.

Reviewer 4 asked for description of the audit teams this was presented in our first revised manuscript on page 5. We have accepted several minor grammatical changes offered by reviewer 4 throughout the new revision of our manuscript.

On page 9 reviewer 4 invited us to add the number of total patients with no visual skin inspection – this we had provided earlier in the first paragraph on page 9. On page 10 we have added deep tissue injury data to the description of severe pressure ulcers and amended the figures presented in page 10 accordingly. We have added the number of verified superficial and severe pressure ulcers in the third paragraph on page 10 as suggested by reviewer 4.

Reviewer 4 noted an apparent inconsistency between the text and Table 2 on Page 11. In the text, it is reported that 155 patients had non-verified pressure ulcers; Table 2 shows non-verified pressure ulcers by anatomical location reporting data on 153 patients and noting that the anatomical location of non-verified pressure ulcers of 2 patients was unreported – so the data in the text and table 2 are consistent.

Reviewer 4 questioned what data was shown in Tables 1 and 2. We have amended the title of Table 2 to make more explicit the distinction between the two tables – Table 1 shows data upon pressure ulcers seen by the audit teams while Table 2 shows data upon pressure ulcers reported to, but not seen by the audit teams.

We have included a new sentence in the Pressure redistributing support surface section to explain what 'appropriate' use of support surfaces was, we thank Reviewer 4 for suggesting this addition.

On page 12 Reviewer 4 asks whether we have an inconsistency between table 3 and the text? The text notes that in 1308 cases data was missing on the mattress allocated to patients, the 1280 total in Table 3 relates to the number of patients with 'no' or 'minimal risk' of developing pressure ulcers. We have amended table 3 to make clearer the explanation of the data shown in the table and added a totals column as suggested by Reviewer 4.

On page 52 Reviewer 4 requested more detail around Table 5 we have amended the table to address this request and provided more detail in the text to allow better interpretation of this data. We have taken the opportunity to correct the denominator of the prevalence figures shown in Table 5 from 5178 (number of patients with a skin assessment) to 8365 (total number of patients included in the audit) and thank Reviewer 4 for suggesting this change.

We trust we have now addressed all the questions posed by the reviewers, and thank all reviewers for helping to strengthen our manuscript.